# Safetxt: a safer sex intervention delivered by mobile phone messaging on sexually transmitted infections (STI) among young people in the UK - protocol for a randomised controlled trial

Caroline Free,[1] Ona L McCarthy [ID],[1] Melissa J Palmer,[1] Rosemary Knight,[2] Phil Edwards [ID],[1] Rebecca French,[3] Paula Baraitser [ID],[4] Ford Colin Ian Hickson,[5] Kaye Wellings,[3] Ian Roberts,[1] Julia V Bailey,[6] Graham Hart,[7] Susan Michie,[8] Tim Clayton,[9] George B Ploubidis,[10] James R Carpenter,[9] Katy M E Turner [ID],[11] Karen Devries [ID],[9] Kimberley Potter[9]

For numbered affiliations see end of article.

**Correspondence to**
Dr Caroline Free;
caroline.free@lshtm.ac.uk

## ABSTRACT

**Introduction** Young people aged 16 to 24 have the highest prevalence of genital chlamydia and gonorrhoea compared with other age groups and re-infection rates following treatment are high. Long-term adverse health effects include subfertility and ectopic pregnancy, particularly among those with repeated infections. We developed the safetxt intervention delivered by text message to reduce sexually transmitted infection (STI) by increasing partner notification, condom use and (STI) testing among young people in the UK.

**Methods and analysis** A single-blind randomised trial to reliably establish the effect of the safetxt intervention on chlamydia and gonorrhoea infection at 1 year. We will recruit 6250 people aged 16 to 24 years who have recently been diagnosed with chlamydia, gonorrhoea or non-specific urethritis from health services in the UK. Participants will be allocated to receive the safetxt intervention (text messages designed to promote safer sexual health behaviours) or to receive the control text messages (monthly messages asking participants about changes in contact details) by an automated remote online randomisation system. The primary outcome will be the cumulative incidence of chlamydia and gonorrhoea infection at 1 year assessed by nucleic acid amplification tests. Secondary outcomes include partner notification, correct treatment of infection, condom use and STI testing prior to sex with new partners.

**Ethics and dissemination** Ethics approval was obtained from NHS Health Research Authority - London – Riverside Research Ethics Committee (REC reference: 15/LO/1665) and the London School of Hygiene & Tropical Medicine. We will submit the results of the trial for publication in peer-reviewed journals.

**Trial registration number** International Standard Randomised Controlled Trials Number: ISRCTN64390461. Registered on 17th March 2016. *WHO trial registration data set* available at: http://apps.who.int/trialsearch/Trial2.aspx?TrialID=ISRCTN64390461.

## Strengths and limitations of this study

► The study design is a large randomised trial to assess the effect of 'safetxt', a safer sex intervention delivered by text message, on objectively measured sexually transmitted infection (chlamydia and gonorrhoea) at 1 year.

► The use of an independent remote computer-based randomisation system linked to an automated message delivery system ensures allocation concealment.

► The secondary outcomes will assess the effect of the intervention on safer sex behaviours: partner notification, condom use and testing before sex with a new partner.

► The process evaluation will explore which intervention components are effective and has the potential to generate general principles to inform similar interventions in the future.

► The inability to blind participants receiving a behavioural intervention is a limitation of the trial, which could introduce bias in the ascertainment of self-reported outcomes.

**Trial protocol version** 12, 19th July 2018.

## INTRODUCTION

Younger people aged 16 to 24 bear the heaviest burden of sexually transmitted infections (STIs) such as chlamydia and gonorrhoea, and their long-term adverse health effects including ectopic pregnancy and subfertility.[1 2] The risk of adverse health effects increases with repeated infections, and re-infection rates following treatment are high: up

to 30% for chlamydia and 12% for gonorrhoea at 1 year.[3] Those with an STI are more likely to acquire further STIs and HIV, if exposed. There are marked inequalities in sexual health; STIs are positively associated with lower educational level[4] and living in more deprived areas.[1 5 6]

Partner notification, condom use and STI testing can reduce infection and re-infection. There is some evidence that existing interventions delivered face-to-face that target condom use and/or STI testing may be effective, but they are limited in their reach or too costly for widespread application.[7] Existing interventions delivered via the media have high reach but their effects have yet to be established.[8] Effective ways to increase partner notification in specialist and primary care settings are needed.[9 10]

Mobile phones have the potential to provide effective, low cost health behaviour support.[11 12] However, evidence for the effect of mobile phone support for safer sex behaviours such as condom use, partner notification and STI testing is equivocal.[13–15] Interventions have targeted testing for STI,[16–21] delaying resumption of sexual activity until 42 days after circumcision[22] and condom use,[17 18 23 24] but none have aimed to increase partner notification. The effect of mobile phone based interventions on STI is not known.

In the UK, 99% of 16 to 24 year olds are mobile phone users (ONS, 2012) and mobile phone ownership is high across all socioeconomic groups. In research leading to this trial, it was demonstrated that interactive support via text message is particularly acceptable in the area of sexual health intervention.[15 25 26]

The safetxt trial builds on the successful intervention development work and pilot trial.[15 26 27] To develop the intervention, we convened a working group including experts in health psychology, contraception research, contraception service provision, information technology and the development of interventions delivered by mobile phone. The intervention messages were developed based on: behaviour change theory, evidence-based behaviour change techniques, the content of effective face-to-face safer sex interventions, the factors known to influence safer sex behaviours, the views of 82 young people collected in focus groups and a questionnaire completed by 100 people aged 16 to 24.[15] The theory and evidence-based intervention is designed to reduce STIs in young people by supporting them in telling a partner about an infection, using condoms and obtaining testing before unprotected sex with a new partner. In a qualitative study with young people, recipients reported that the tone, language, content and frequency of messages was appropriate.[26] Messages reportedly increased knowledge and confidence in how to use condoms and reduced stigma enabling them to tell a partner about an STI. Sharing messages with their partner enabled participants to negotiate condom use.

The pilot trial demonstrated that methods of recruitment, randomisation, intervention delivery and follow-up were successful and that a full-scale randomised controlled trial of the safetxt intervention is feasible.[27]

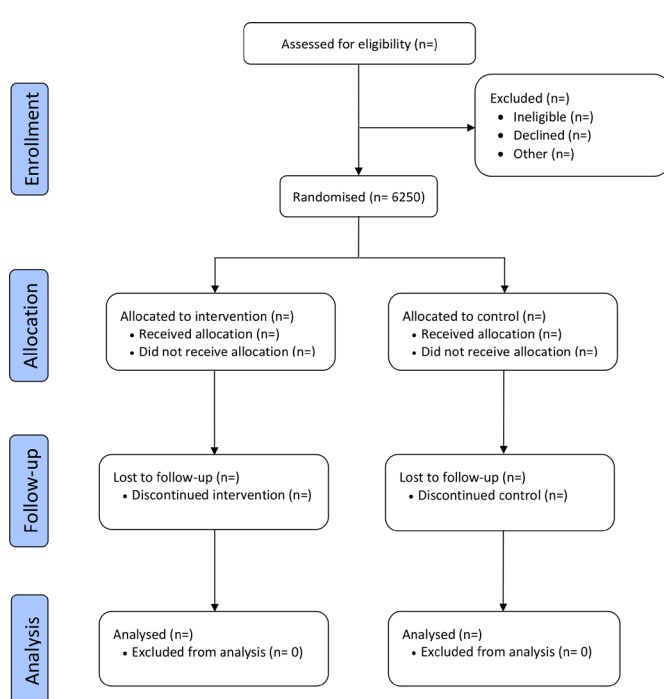

**Figure 1** Trial flow chart.

## Objectives

The primary objective of this trial is to quantify the effects of the safetxt intervention compared with a control group receiving usual care and messages about trial participation on chlamydia or gonorrhoea infection at 1 year. Secondary objectives are to determine the effects of safetxt on partner notification and condom use at 4 weeks and on condom use and STI testing at 1 year. Which intervention components are effective will be explored by collecting data on the theoretical constructs on the pathway to behaviour change influenced by the intervention components. The cost-effectiveness of the intervention will be established.

## METHODS AND ANALYSIS
### Design and setting

Safetxt is a single-blind parallel group randomised superiority trial with a 1:1 allocation ratio designed to establish the effects of a safer sex intervention delivered by text message on the cumulative incidence of chlamydia and gonorrhoea infection at 1 year. (See figure 1 for the trial flow chart).

Potential participants testing positive for chlamydia, gonorrhoea or diagnosed with non-specific urethritis (NSU) will be identified from UK STI testing services by research staff based at the service (a full list of participating services is available at http://safetxt.lshtm.ac.uk/participating-sites/). The intervention is delivered by text message in the community. Trial recruitment started on the 1st April 2016. Completion of follow-up and closure of the data set is planned for early spring 2020.

## Characteristics of participants

### Inclusion criteria

Participants will be eligible if they are between 16 and 24 years of age, own a personal mobile phone, are able to provide informed consent and who, in the 2 weeks prior to their recruitment, have either been diagnosed with chlamydia, gonorrhoea or NSU or have started treatment for chlamydia, gonorrhoea or NSU.

### Exclusion criteria

Participants will be excluded if they are known to be a sexual partner of someone already recruited to the trial.

### Recruitment and consent

Research staff based at STI testing services will identify potential participants. They will provide potential participants with trial information at one of three time points, that is, when potential participants (i) attend the service and are diagnosed with chlamydia, gonorrhoea or NSU; (ii) receive positive test results for chlamydia or gonorrhoea by phone or (iii) collect treatment for chlamydia, gonorrhoea or NSU from services.

Research staff will provide potential participants with verbal and written information (online supplementary file 1) about the trial and/or participants will view trial information on the website. Participants will be able to join the trial by either providing informed written consent to the research staff, or by providing consent online (online supplementary file 2) at the trial website or site staff can ask eligible participants if they are happy for their details to be given to research assistants based at the London School of Hygiene & Tropical Medicine Clinical Trials Unit (LSHTM CTU). In this case, LSHTM CTU will contact participants to recruit to the trial. Strategies for achieving adequate enrolment include site visits, training, feedback and rewards, sharing successful recruiting strategies and newsletters.

### Allocation and blinding

Participants will be randomly allocated (1:1 ratio), using a computer-based randomisation system, to a safer sex intervention delivered by text messaging, or to a control group, ensuring allocation concealment. Allocation concealment is ensured as the randomisation system is independent, automated and remote.

An electronic link to the computer-based randomisation programme will result in the generation of a research number and allocation to the intervention or control group. The system will then automatically deliver intervention or control group texts according to the allocation.

Given the nature of the intervention, participants will be aware of their assigned group. Research assistants entering data, laboratory staff and researchers conducting the final analysis are masked to allocation until after the analysis is complete.

### Interventions

#### Safetxt intervention group

The intervention aims to increase safer sex in three ways: (i) encouraging participants to correctly follow STI treatment instructions including informing partner(s) about infection, (ii) promoting condom use with new or casual partners and (iii) encouraging participants to obtain testing for STI prior to unprotected sex. Participants in the intervention group will receive regular messages delivered by text message in community settings according to a predetermined schedule. The intervention was informed by the capability, opportunity and motivation model of behaviour.[28] It aims to influence the knowledge, beliefs, self-efficacy, skills, social and interpersonal influences that have important effects on motivation, capability and opportunity to reduce sexual risk behaviour.

Over the first 10 days participants are sent messages targeting engagement with the intervention, taking treatment, avoiding sex for 7 days after treatment and telling partner(s) about an infection. These messages provide non-judgemental, non-stigmatising information about STI. They provide suggestions about when, where and how to tell partner(s) about an infection and examples of how others told partners covering a range of different types of relationship (eg, casual, long-term).

Messages then target condom use and testing for STIs before having sex without a condom with a new partner. Topics cover risk assessment, instructions on how to use condoms, positive aspects of condom use, tips on preventing condom problems and examples of how others resolved condom use problems. Participants are prompted to think about their own success in achieving safer sex strategies, risks they had taken and what they could do differently in the future. Messages included advice regarding testing before unprotected sex with a new partner. Participants are sent links to: services, support for those concerned about partner violence, and web-based information regarding contraception, alcohol and sexual risk, how to use a condom and general communication about sex. The messages provide social support for safer sex behaviours and acknowledge participants experiences.

The intervention employs educational, enabling and incentivising behaviour change functions and 12 behaviour change techniques: information about health consequences of behaviour, instruction on how to carry out the behaviour, demonstrations of risk reduction behaviour, social support, emotional support, social rewards, non-specific incentives, encouragement to add objects to the environment, anticipated regret, problem solving, action planning techniques and reframing.[29] The information on safer sexual practices is in accordance with existing guidelines.

The messages are tailored according to gender and sexual orientation. Women who have sex with men only (WSM), men who have sex with men only (MSM), men who have sex with men and women (MSMW), women who have sex with men and women (WSMW) are sent messages about how others had negotiated condom use. WSM and men who have sex with women only (MSW) are sent messages about emergency contraception. MSM and

MSMW are sent messages about post exposure prophylaxis. Women who have sex with women only (WSW) are not sent messages about condom use. The information provided is specific to the STI diagnosed. This tailoring results in different numbers of messages being sent to those of different gender and sexual orientation.

The core message sets include: 74 messages for WSM and WSMW, 42 messages for WSW, 69 messages for MSW and 79 messages for MSMW and 76 messages for MSM. Recipients can request additional messages on specific topics. Participants are sent: four messages per day for days 1 to 3, then one to two messages per day for days 4 to 28, then two to three messages per week for month 2 and two to five messages per month for months 3 to 12.

### Control group

Participants in the control group will receive a monthly untailored text message asking for information about changes in postal or email addresses. Example control group message: 'Thank you for taking part in the texting study. Remember to let us know if your contact details have changed by replying to this text or emailing safetxt@lshtm.ac.uk'.

All participants will receive usual care and will be free to seek any other existing service or support they wish. The control group is not attention matched as during the pilot work young people reported that it was irritating to receive the same number of messages as the intervention group about another health topic, when they had just been diagnosed with an STI. If the trial shows a benefit it is possible that this could be due to simply receiving the number of messages involved in the intervention, rather than the message content. However, given the lack of effect of some mobile phone message based interventions in sexual health this seems unlikely.[14]

All messages are sent automatically from a large database to an aggregator, which conveys messages to each participant in the community via their network. The success of delivery of messages at each step is monitored by the aggregator and computer system that generates the messages. A member of the trial team will automatically be notified if there is any failure in the delivery of messages. All participants will be able to set embargoed times when they do not want to receive messages. Participants will be able to stop text messages, but continue with the trial follow-up, by texting 'stop' to the short code number.

### Outcomes

Our primary outcome will be the cumulative incidence of chlamydia and gonorrhoea infection at 1 year assessed by nucleic acid amplification tests: urine for men (with additional pharyngeal and anal swabs for MSM) and self-taken vulvovaginal swab for women.

### Secondary outcomes
#### Secondary outcomes at 4 weeks
Behaviours

- Whether participants took the (prescribed antibiotic) treatment and avoided sex for 7 days after treatment.
- Whether they told the last person they had sex with before testing positive, that they needed to get treatment.
- Clinic attendance by partner for treatment (identified from clinic records).
- Condom use at last sex.

Process outcomes – scores of the theoretical constructs underlying the components of the intervention (behaviour change mediators):

- Attitudes towards partner notification.
- Self-efficacy in telling a partner about an infection.
- Self-efficacy in negotiating condom use.
- Correct condom use self-efficacy.
- Knowledge related to STIs.

### Secondary outcomes at 1 year
STIs

- Diagnosed with any STI after joining the trial according to self-report confirmed by postal test results and clinic records.

Behaviours

- Condom use at last sex.
- Number of sexual partners since joining the trial.
- Sex with someone new since joining the trial.
- Condom use at first sex with most recent new partner.
- Self-reported sexually transmitted infection testing for self - prior to first sex with most recent new partner.
- Sexually transmitted infection testing for self - prior to first sex with most recent new partner (confirmed according to clinic record that testing occurred).
- Participant's report as to whether their most recent new partner was tested for sexually transmitted infection prior to sex with them.
- Number of sexual partners since joining the trial.
- Reading and sharing of intervention content.
- Number of text messages read.
- Whether anyone else read the messages, if yes how did the participant feel about them reading the messages?
- Reading someone else's messages in the trial (to be reported for control group participants).
- Someone else in the trial reading participants' messages (to be reported for intervention group participants).

Potential harms

- Car accident where the participant was the driver in the past year.[30]
- Experience of partner violence in the past year.[31]

### Data collection

The primary outcome will be assessed using chlamydia and gonorrhoea tests collected by post at 1 year and clinic records of completed tests. STI test kits for chlamydia and gonorrhoea will be posted (in P650 standard packaging) to respondents. Directions in the pack will ask participants to provide a vaginal swab (women), a urine sample (men) and additional anal and pharyngeal samples (men

| | STUDY PERIOD | | | | |
|---|---|---|---|---|---|
| | Enrolment | Allocation | Post-allocation | | Close-out |
| **TIMEPOINT**** | **0** | **0** | *Wk 4* | *Wk 52* | *Wk 76* |
| **ENROLMENT:** | | | | | |
| **Eligibility screen** | X | | | | |
| **Informed consent** | X | | | | |
| **Allocation** | | X | | | |
| **INTERVENTIONS:** | | | | | |
| *safetxt text messages* | | | ←————————→ | | |
| *control text messages* | | | ←————————→ | | |
| **ASSESSMENTS:** | | | | | |
| *Socio-demographic characteristics* | X | | | | |
| *Recent sexual behaviour* | X | | X | X | X |
| *STI diagnosis* | X | | | | |
| *Correct treatment of STI* | | | X | | |
| *Partner notification* | | | X | | |
| *Attitudes, self-efficacy, knowledge* | | | X | | |
| *STI test (postal NAAT)* | | | | X | X |
| *STI testing prior to sex with new partner* | | | | X | X |
| *Number of text messages read* | | | | X | X |
| *Anyone else read text messages* | | | | X | X |
| *Experience of car accident* | | | | X | X |
| *Experience of partner violence* | | | | X | X |

**Figure 2** Schedule of enrolment, interventions and assessments. NAAT,nucleic acid amplification test; STI, sexually transmitted infections.

who have sex with men) and post it in the prepaid and addressed envelope to the laboratory. Test kits will be identified by lab number only, rendering the laboratory staff blind to the participant's allocation. The accredited laboratory will destroy samples after testing is completed.

Results of the STI testing will be reported on the secure trial lab site by lab code only. Clinical records will be checked to identify participants who have had a positive STI diagnosis since joining the study, confirm self-reported STI testing and partner attendance for treatment.

At 4 weeks and at 1 year, postal questionnaires will be sent to all participants to collect self-reported data. Non-responders will be contacted by any method the participant agrees to at enrolment (post, email, text message, telephone call). Participants can directly enter self-reported outcome data via a web-based data entry form. Paper based self-reported outcome data or self-reported data collected by telephone will be directly entered into the web-based data entry form by a trial assistant blinded to treatment allocation.

Participants will be sent a £5 unconditional incentive with each postal request, that is, when sending the 4 weeks questionnaire and 1 year test and questionnaire. Participants who return the test sample will be sent £20. See figure 2 for the Schedule of enrolment, interventions and assessments. All outcome data will be collected on participants who choose to stop messages unless they withdraw from the trial.

## Data management

Anonymised research data will be held on a secure system and password protected. All data systems will be set up with checks to alert the trial assistants if data being entered are illogical, inconsistent or incomplete.

Personal details will be stored on a password protected computer held on a secure server at the LSHTM. In accordance with the Data Protection Act of 1998,[32] this information will be stored separately from any anonymised research data, and will be deleted at the end of the study.

Any paper-based data will be locked in a cabinet within a room. All trial procedures are in accordance with the principles of Good Clinical Practice. Essential documents of the sponsor/trial organisers and investigators will be retained for at least 10 years after completion of the trial.

## Sample size

The study is powered for the primary outcome measure of cumulative incidence of chlamydia and/or gonorrhoea infection at 1 year. Two main factors determine the number of participants needed for this trial: the estimated event rate, and the size of the treatment effect. Our estimates are based on the following data:

### Estimated event rate

The estimated event rate for the cumulative incidence of STI at 1 year is 20%, based on the event rate in cohort studies and the pilot trial.[3]

### Size of treatment effect

Because the intervention can be administered to large populations at low cost, even a modest reduction in treatable STIs would be worthwhile. The trial has therefore been designed to detect a reduction in chlamydia or gonorrhoea infection of 20% versus 16% (relative risk 0.8), which is similar to the effects of face-to-face safer sex interventions.[7]

### Sample size calculation

In the pilot trial 2% of the control group viewed messages delivered to intervention participants. If the real difference in STI infection at 1 year follow-up is 20% versus 16% then with contamination of 2% the trial would detect a difference of 19.9% versus 16%. (Calculated based on 2% of the control group having an infection rate the same as the intervention group = (98% x 20%) + (2% x 16%) = 19.9%).

Five thousand participants would provide 90% power to detect this difference, allowing for 20% losses to follow-up.

The Trial Steering Committee (TSC) reviewed the blinded event rate after 546 patients had completed 12 months follow-up and recommended an increase in the sample size to 6250 due to a lower than expected event rate of 15.6%.

## Analysis

A detailed statistical analysis plan (SAP) will be finalised and submitted to the TSC before the end of the trial and

unblinding. For the primary outcome and other binary outcomes, we will estimate any differences between the groups using logistic regression and we will report ORs with 95% CIs and p values, adjusting for key baseline predictors of outcome. These will be specified in advance in the SAP. For continuous outcomes, we will use linear regression to test for a difference in mean scores between the arms. All analyses will be based on the intention-to-treat principle.

For the primary analysis Multiple Imputation by Chained Equations (MICE) will be used in order to account for missing data.[33] MICE makes appropriate assumptions for accommodating missing data in the analysis based on the predictors of outcome and the predictors of loss to follow-up. MICE is recognised as a way of reducing bias and increasing precision of trial results and is increasingly used as the primary analysis in randomised controlled trials.[12 33] One hundred imputed data sets will be generated and the point estimates and SEs will be combined using Rubin's rules. We will perform information-anchored sensitivity analysis using the delta-method.[34]

### Supplementary analyses

Before fitting our primary analysis model, we need to identify appropriate auxiliary variables to include in the imputation model. Such auxiliary variables should have information about the missing outcome. Specifically, an independent statistician will use a data set without the treatment variable and perform regression analyses to identify key predictors of outcomes and which combination of baseline variables and time to negative STI test in clinic should be included as auxiliary variables in the imputation model. We will supplement this by performing and reporting a complete case analysis (where any participants with missing information on any covariate or outcome shall be excluded).

### Subgroup analyses

A limited number of subgroup analyses will be undertaken adjusting for other key predictors of outcome. Subgroup analyses include: age,[16–24] sexual orientation (men who have sex with men or men and women, men who have sex only with women, women who have sex with men or men and women, women who only have sex with women), ethnic group (Caucasian, Black, other) and age at which left education (16 or under, 17 or over). The subgroups will be analysed by inclusion of an interaction term between treatment group and the subgroup in the appropriate regression model for the outcome. Data will be presented by categories of subgroup using effect estimates and 99% CIs. For each subgroup analysis, missing outcomes will be imputed consistent with the hypothesis of an interaction.[35]

### Process outcomes

Candidate questionnaire items were identified for the measurement of the following constructs: attitudes towards partner notification, self-efficacy in telling a partner about an infection, self-efficacy in negotiating condom use, correct condom use self-efficacy, knowledge related to STIs. At 4 weeks, each of the five process outcomes were measured using the candidate questionnaire items (three to five items per process outcome). Based on these items, we will carry out confirmatory factor analysis to refine and identify a valid measurement model. For the main trial analyses, the final measurement model will be combined with the intervention group allocation, along with key baseline predictors, extending the measurement model to a Multiple Causes Multiple Indicators model. This model will be used to estimate the effects of the intervention on the five hypothesised process outcomes.

### Economic evaluation

The economic modelling required to assess the cost-effectiveness of the intervention will estimate the annual probability of members of the target group acquiring chlamydia, gonorrhoea and NSU with and without the intervention (based on the experience of those in the control and intervention arms of the trial). Detailed information will be collected on the costs of delivering the intervention. Secondary sources will be used to estimate the future National Health Service (NHS) costs avoided as a consequence of avoided infections.

### Patient involvement

Patients were involved in the development of the intervention[15] and the design of the trial materials and follow-up procedures.[36] A patient is also a member of the Trial Steering Committee. Young people will be involved in the dissemination of the trial results.

### Monitoring

The safetxt trial will not have a separate Data Monitoring Committee. No interim analyses of intervention effects will be conducted since the trial is of a behavioural support intervention unlikely to cause harm, and therefore, there are no stopping rules. Analysis of intervention effects will be conducted once at the end of the trial. The sponsor may audit the trial per their own risk assessment and schedule.

Harms will be assessed by self-reported data. Car accidents are the only demonstrated harm resulting from text messaging, hence the intention to collect data regarding involvement in road traffic accidents. The safetxt intervention aims to increase partner notification of STI status. Fear of partner violence has been reported to be a barrier to partner notification and partner notification has been identified as a factor which may trigger partner violence. However, no randomised controlled trials targeting increased partner notification detected a difference in partner violence between the control intervention groups. Data will be collected regarding the experience of partner violence at 12 months.

In addition, participants will be asked an open-ended question in the 12-month questionnaire: 'Did anything

good or bad happen as a result of being involved in the study or receiving the text messages? Please describe.' The Clinical Trials Unit will keep detailed records of all adverse events reported. Reports will be reviewed by the Principal Investigator and the TSC to consider intensity, causality and expectedness. As appropriate, these will be reported to the sponsor and the Ethics Committee.

## Ethics and dissemination

All trial participants will provide written informed consent prior to enrolment in the trial. Participants' can withdraw from the trial on request. Participant will receive usual NHS care there is no ancillary care.

Any modifications to the protocol will be approved by the ethics committee prior to implementation. Records of any important modifications will be submitted as an addendum in the journal in which this protocol is published.

Trial results will be published open-access in peer-review journals. Authorship will be on the basis of meeting the criteria recommended by the International Committee of Medical Journal Editors.

After the publication of our main trial findings, the anonymised trial data set will be available on request from the corresponding author.

## DISCUSSION

STIs such as chlamydia and gonorrhoea confer a heavy burden of disease among young people with long-term sequelae such as infertility. Safer sex behaviours such as condom use, notifying partner(s) about an existing STI and STI testing reduce the risk of STI, but young people may lack the knowledge, confidence and skills needed to adopt these behaviours. Safer sex interventions delivered by mobile phone are acceptable to young people and show promise in increasing safer sex behaviours, but to date their effects on STI are not reliably known.

If it proves to be effective, this low-cost intervention could have an important impact in improving sexual health and reducing inequalities among young people in the UK. There is likely to be international interest in the impact of the intervention as short written messages delivered via mobile phones are increasingly used for behavioural support worldwide and sexually transmitted infections remain an important cause of morbidity and mortality.

**Author affiliations**
[1]Population Health, London School of Hygiene and Tropical Medicine, London, UK
[2]Clinical Trials Unit, MSD, London School of Hygiene and Tropical Medicine, London, UK
[3]Social and Environmental Health Research, London School of Hygiene and Tropical Medicine, London, UK
[4]Centre for Global Health, King's College London, London, UK
[5]Sigma Research, London School of Hygiene and Tropical Medicine, London, UK
[6]Primary Care and Population Health, University College London, London, UK
[7]Department of Infection and Population Health, University College London, London, UK
[8]Centre for Outcomes Research and Effectivenes, University College London, London, UK
[9]Department of Medical Statistics, London School of Hygiene and Tropical Medicine, London, UK
[10]Department of Social Science, University College London Institute of Education, London, UK
[11]Bristol Vetinary School, University of Bristol, Bristol, UK

**Collaborators** The Trial Steering Committee (TSC) may recommend changes to the overall trial management processes for which there is evidence the current process may compromise the conduct of the trial or the safety of participants. The members of the TSC are: Professor Pippa Oakeshott (Chair), Dr Andrew Copas, Professor Michael Ussher, Mr Colum McGrady, Dr Michael Brady and Professor Caroline Free.

**Contributors** CF conceived the study, contributed to the development of the intervention and statistical analysis plan, is a member of the Trial Management Group and had overall responsibility for the design and conduct of the trial and writing the protocol. OM contributed to the development of the intervention, contributed to the statistical analysis plan, is a member of the Trial Management Group, contributed to the trial design, the refinement of the study protocol and approved the manuscript. MP contributed to the statistical analysis plan, is a member of the Trial Management Group, contributed to the process evaluation design, contributed to the trial design, the refinement of the study protocol and approved the manuscript. RK is a member of the Trial Management Group, is responsible for the day-to-day management of the trial, contributed to the trial design, the refinement of the study protocol and approved the manuscript. PE is the trial statistician who provides general statistical oversight, contributed to the statistical analysis plan, contributed to the trial design, the refinement of the study protocol and approved the manuscript. RF contributed to the development of the intervention, provided input on the trial outcome measures, contributed to the trial design, the refinement of the study protocol and approved the manuscript. PB contributed to the development of the intervention, contributed to the trial design, the refinement of the study protocol and approved the manuscript. FH adapted the intervention messages for men who have sex with men, contributed to the trial design, the refinement of the study protocol and approved the manuscript. KW contributed to the development of the intervention, provided input on the trial outcome measures, contributed to the trial design, the refinement of the study protocol and approved the manuscript. IR provided expertise on trial design, contributed to the refinement of the study protocol and approved the manuscript. JB contributed to the development of the intervention, the trial design, the refinement of the study protocol and approved the manuscript. GH provided fundamental intellectual input into the design of the trial, the refinement of the study protocol and approved the manuscript. SM provided behavioural science expertise, which informed the development of the intervention, contributed to the trial design, the refinement of the study protocol and approved the manuscript. TC provided statistical expertise regarding the supplementary and subgroup analyses, provided general statistical advice, contributed to the trial design, the refinement of the study protocol and approved the manuscript. GP provided statistical expertise for the Generalised Latent Variable Modelling framework, contributed to the trial design, the refinement of the study protocol and approved the manuscript. JC provided statistical expertise regarding multiple imputation, contributed to the trial design, the refinement of the study protocol and approved the manuscript. KT leads the economic evaluation, contributed to the trial design, the refinement of the study protocol and approved the manuscript. KD contributed to the development of the intervention, contributed to the trial design, the refinement of the study protocol and approved the manuscript. KP is a member of the Trial Management Group, is responsible for the day-to-day management of the trial, contributed to the trial design, the refinement of the study protocol and approved the manuscript.

**Funding** This trial is funded by the National Institute for Health Research Public Health Research (NIHR PHR) Programme (project number 14/182/07). London School of Hygiene & Tropical Medicine is the main research sponsor for this trial. For further information regarding the sponsorship conditions, please contact the Research Governance and Integrity Office: London School of Hygiene & Tropical Medicine, Keppel Street, London, WC1E 7HT Tel: +44 207 927 2626 Email: patricia.henley@lshtm.ac.uk.

**Disclaimer** The funding body has no role in any aspect of the design and conduct of this study, or the decision to submit the report for publication. The sponsor had no role in any aspect of the design and conduct of this study, or the decision to submit the report for publication.

**Competing interests** None declared.

**Patient consent for publication** Not required.

**Ethics approval** Ethics approval for this trial was provided by the NHS Health Research Authority - London – Riverside Research Ethics Committee (REC reference: 15/LO/1665) and the London School of Hygiene & Tropical Medicine.

**Provenance and peer review** Not commissioned; externally peer reviewed.

**ORCID iDs**
Ona L McCarthy http://orcid.org/0000-0002-9902-6248
Phil Edwards http://orcid.org/0000-0003-4431-8822
Paula Baraitser http://orcid.org/0000-0002-3354-6494
Katy M E Turner http://orcid.org/0000-0002-8152-6017
Karen Devries http://orcid.org/0000-0001-8935-2181

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
