## [Reviewer comments · BMJ Open]

ARTICLE DETAILS

TITLE (PROVISIONAL)	safetxt: a safer sex intervention delivered by mobile phone messaging on sexually transmitted infections (STI) among young people in the UK- protocol for a randomised controlled trial
AUTHORS	Free, Caroline; McCarthy, Ona; Palmer, Melissa; Knight, Rosemary; Edwards, Phil; French, Rebecca; Baraitser, Paula; Hickson, Ford; Wellings, Kaye; Roberts, Ian; Bailey, Julia; Hart, Graham; Michie, Susan; Clayton, Tim; Ploubidis, George; Carpenter, James; Turner, Katy; Devries, Karen; Potter, Kimberley

VERSION 1 – REVIEW

REVIEWER	Catherine M Lowndes Public Health England United Kingdom
REVIEW RETURNED	12-Aug-2019

GENERAL COMMENTS	I had to do fair amount of digging to understand the manuscript, including previous publications and trial registration documentation, mainly because there were no dates given in the protocol. The trial was registered and initiated in 2016, and took nearly 3 years to complete. According to both trial sites, recruitment has been completed. The pilot data for the trial has been published, including as an HTA assessment. So while there are no trial outcome data in the manuscript (one of BMJ's criteria for publication), data collection is finished, as per the end of 2019. Please see attached doc for details. As such, I do not think it will add to the already-published body of work, particularly since no major changes were made to the protocol as a result of the pilot. The only change noted in the present manuscript, made after unmasking by the DMC, was an increase in sample size from 5,000 to ~6,750. In addition, in its current state - particularly the lack of dates, the manuscript is confusing, and I had to read several reports and papers to understand exactly what happened when. At the least a table of all relevant dates would have to be added, although again this would add nothing to the body of knowledge.
---

REVIEWER	Judith B. Cornelius UNC Charlotte United States
REVIEW RETURNED	15-Aug-2019

GENERAL COMMENTS	I do not see dates of the study it is not clear why you chose to send the number of text messages
--

	that you sent and why the numbers vary per participants, 79, 76 etc- further clarification is requested for this information How can you verify if others read the messages? The control group is not a attention control group? Any rationale as to why?
--	---

REVIEWER	Guangyu Tong Duke University, USA
REVIEW RETURNED	17-Oct-2019

GENERAL COMMENTS	The present trial studies the “safetxt” intervention on sexually transmitted infections among young people aged 16-24. The authors wrote a very comprehensive protocol covering all aspects from recruitment to analysis of good details. In below, I provide 9 comments to help improve this protocol. 1. Recruitment: a. On p.6-7, the inclusion criteria say “have either been diagnosed with chlamydia, gonorrhea, or NSU in the last 2 weeks, or have started treatment for Chlamydia, gonorrhea or NSU in the last two weeks.” While it seems these diseases might be prevalent enough at the trial sites, is it feasible to recruit all the 6250 people at the same time? The condition “in the last 2 weeks” sounds pretty restrictive. If not, are the subjects sequentially enrolled? Does that influence the estimation of the treatment effect?) b. While it might be difficult to offer more information about the trial sites at this stage, some description/consideration on this could be crucial because the choice of trial sites influences the outcomes of this study. Are the trial sites geographically clustered? A geomap might be helpful. For instance, if the randomization is performed among gay men diagnosed with chlamydia and gonorrhea from the same hospital, (due to their much smaller sexual networks) the contamination rate between the treatment and control groups could be high after one year, which may threaten the validity of the inference. Also, based on http://safetxt.lshtm.ac.uk/participating-sites/, there are at least 40-50 sites and is there any consideration for “clustered randomization”? Is it better to consider “cluster randomization” so that the between-group contamination can be reduced? c. Population heterogeneity: The large sample size is a great advantage of this trial. The authors are also aware of the different sexual orientations of potential subjects. On p.12, is it possible to also give some group-specific calculations (e.g., MSW, WSW, MSM) for sample sizes? It is potentially a great contribution of this trial to have group-specific estimates of the treatment effect. 2. Outcomes: a. On p. 10 the authors listed all the secondary outcomes at four weeks and one year. One reason for these outcomes being secondary, I guess, is that they are self-reported. While some of these outcomes are immediately relevant, the outcomes such as “car accident where the participant was a driver in the past year”, and “experience of partner violence in the past year” are not immediately relevant to me. Though secondary, why do these outcomes matter? b. Also, on p.10. I am curious about what the outcome of “contamination between intervention and control group” means. It sounds like this is some information for the statistical inference of
---

	the treatment effect rather than an outcome. c. Is there any way to categorize the outcomes so that they can be better organized in the paper? For instance, “Diagnosed with any STI after joining the trial...” and “Self-reported sexually transmitted infection test for self...” sound like similar types of outcomes. Rather than providing a laundry list of them, putting them together under the category of sth. like, “self-reported testing results”, will make it much easier for people to understand the differences between them and have a better sense of the design of this trial. 3. Interventions: a. On p.11, why are there only 42 messages for WSW but 79 messages for MSW? Based on the following description on p.11, “Participants are sent 4 messages per day for days 1-3, then 1-2 messages per day for days 4-28, then 2-3 messages per week for month 2, and 2-5 messages per month for months 3-12”, The minimum number of messages one will receive is $4*3+1*28+2*4+2*10=68$. Then how comes WSW only receives 42? Are they treated differently? b. The meaning of the “control message” is unclear. For instance, are they sent at the same frequency as the “treatment message”? Are they also tailored to people with different sexual orientations? Also, what is the content of the message? Is it information only about general care or also STD related, but more general? Clarifying the “control message” can be quite important here, because it will let the reader know what the intervention/treatment effect means in this study. i.e., is the intervention about the message, about message “being tailored”, or about the “STD” content of the message? 4. Data analysis a. The authors offer a plan for data analysis, which also covers the potential scenarios for handling missing data with MICE. On p.13, it says MICE is based on “predictors of the outcome” and “predictors of loss to follow up.” So, are they the same group of predictors? Is it talking about study covariates? If not, what are examples of “predictors of loss to follow up” other than “predictors of the outcome”? Also, by principle, the number of imputations should be based on the percentage of missingness, and it gains no more precision to impute 100 datasets when only 15% of data are missing (15 is probably enough).
--	---

VERSION 1 – AUTHOR RESPONSE

Reviewer: 1.

Reviewer 1 reported that they found the manuscript hard to understand partly because there were no dates given in the protocol. They concluded that trial recruitment and follow up were completed and wanted to know why outcome data were not included.

We apologise for not including the trial start date and anticipated end date. We have now included the dates for the start of recruitment (1/4/2016) and anticipated timing for completion of follow up and closing the data set (early spring 2020). We confirm that trial recruitment is completed. Trial follow up is ongoing. We would like to clarify that we have not reported any outcome data as the trial is still ongoing and this is a trial protocol.

Reviewer 1 did not think that the protocol would add to the already published work as they considered

that no major changes were made to the protocol as a result of the pilot trial, other than the main trial sample size calculation.

It is considered good practice to publish trial protocols. We would like to clarify that there are important differences between the pilot work and this trial protocol:

Eligibility criteria.

In the previous published pilot work the eligibility criteria included those diagnosed with chlamydia, and people reporting unsafe sex (defined as more than 1 partner and at least 1 occasion of sex without a condom in the last year) (1, 2).

The substantive trial includes those diagnosed with chlamydia, gonorrhoea or non-specific urethritis (NSU) (or on treatment for these), but does not include those reporting unsafe sex without a sexually transmitted infection (STI) diagnosis/treatment (1, 2).

Pilot inclusion criteria:

'Residents of England aged 16–24 who had received either a positive chlamydia test result or reported unsafe sex in the last year (defined as more than one partner and at least one occasion of sex without a condom), were literate in English and who owned a personal mobile phone were eligible.' (2)

In this trial the inclusion criteria are:

'Participants will be eligible if they are between 16 and 24 years of age, own a personal mobile phone, are able to provide informed consent, and have either been diagnosed with chlamydia, gonorrhoea, or NSU in the last 2 weeks, or have started treatment for Chlamydia, gonorrhoea or NSU in the last two weeks.' (protocol)

Outcomes: In the pilot trial the primary outcomes were recruitment and follow up rates. The other key outcomes for the feasibility work conducted prior to this main trial were participants' views and experiences of the acceptability of the intervention (See abstract, ref 1).

The primary outcome for the main trial is the cumulative incidence of chlamydia and gonorrhoea infection at one year. The secondary outcomes are detailed in the protocol submitted and cover behavioural outcomes, process outcomes, reading of messages and outcomes linked to safety.

Analysis: The submitted main trial protocol includes the plans for analysis of the trial data including the use of logistic regression and Multiple Imputation by Chained Equations (MICE) to account for missing data. Details regarding supplementary analyses and subgroup analyses are provided. These were not part of the pilot trial data analysis plan.

Plans for a health economic analysis are outlined in the submitted trial protocol. These were not included in the pilot trial.

Reviewer 2.

Asked to see dates of the study.

Thank you. These have now been included.

Asked why the numbers of text messages vary per participants.

Thank you for your query. The number of messages sent varies according to gender and sexual orientation as described in the section in the intervention description regarding tailoring.

To make this clear we have moved the order of the content for the intervention description so that information about the number of messages sent appears straight after the section on tailoring.

To clarify this further, we have added a sentence to the end of the section about tailoring of message sets according to gender and sexual orientation stating. 'This tailoring results in different numbers of messages being sent to those of different gender and sexual orientation.'

How can you verify if others read the messages?

It is not possible for us to verify if other people looked at participants' phones and read their text messages. We are only able to ask participants whether anyone read their messages. Participants will only be able to tell us about other people looking at their phones and messages that they are

aware of.

The control group is not an attention control group? Any rationale as to why?

The control group is not attention matched as all participants have very recently been diagnosed with an STI and in the pilot work potential participants reported that receiving a similar number of messages to the intervention group about another topic at this time would be irritating and annoying. We will discuss the limitations of a non-attention matched control in our discussion of the main trial results. All participants do have access to the counselling available as part of usual care in clinical settings.

Reviewer: 3

The authors wrote a very comprehensive protocol covering all aspects from recruitment to analysis of good details. Reviewer 3 asked for some clarifications:

Recruitment:

Reviewer 3 wanted to know if we were recruiting all participants at the same time or sequentially. We would like to confirm that the participants are recruited sequentially. Each individual has in the 2 weeks prior to their recruitment, either been diagnosed with chlamydia, gonorrhoea, or NSU or has started treatment for Chlamydia, gonorrhoea or NSU.

To make this clearer we have altered 'last 2 weeks' to 'in the 2 weeks prior to their recruitment'. 'Participants will be eligible if they are between 16 and 24 years of age, own a personal mobile phone, are able to provide informed consent and who, in the 2 weeks prior to their recruitment, have either been diagnosed with chlamydia, gonorrhoea, or NSU, or have started treatment for Chlamydia, gonorrhoea or NSU.'

Reviewer 3 asked for further information about recruitment sites and the consideration of clustering. Thank you for your thoughtful comment. We gave clustering some considerable thought in the design phase. We can confirm that the 52 recruitment sites are across England with one site in Scotland. We measured contamination across intervention/control groups (sharing messages between participants) in the pilot phase in three sites, when it was found to be low. Participants known to be a partner of someone in the study are excluded. We agree that some contamination might still occur and considered the potential advantages and disadvantages of a cluster RCT design (3). We opted for individual level randomisation as cluster trials can be susceptible to recruitment bias, in the pilot phase levels of contamination were low, and we have allowed for some contamination by increasing our sample size. In this trial, all participants have been diagnosed with a STI. This is a private matter and so in this trial sexual health education materials would be less likely to be shared than general sex education messages delivered to general populations of 16-24 year olds. If contamination occurred, it would reduce the effect estimate identified in the trial.

The reviewer asked if it was possible to provide group specific calculations for sample sizes.

We can confirm that the trial is not designed to look at the effect estimates for the primary outcome in sub groups. After the trial additional secondary analyses could possibly be conducted exploring effects and how much power there is for behavioural outcomes in the different subgroups.

Outcomes: The reviewer asked for clarification as to why car accidents or partner violence are important outcomes.

These outcomes are possible harms or negative effects that could plausibly be caused by the intervention. To clarify this we have provided further information in the trial protocol:

'Harms will be assessed by self-reported data. Car accidents are the only demonstrated harm resulting from text messaging, hence the intention to collect data regarding involvement in road traffic accidents. The safetxt intervention aims to increase partner notification of STI status. Fear of partner

violence has been reported to be a barrier to partner notification and partner notification has been identified as a factor, which may trigger partner violence. However, no randomised controlled trials targeting increased partner notification detected a difference in partner violence between the control intervention groups. Data will be collected regarding the experience of partner violence at 12 months.'

The reviewer asked about the measurement of contamination as an outcome:

In view of the possibility that intervention group participants might share messages with a control group participant, we will collect data about viewing of other trial participants messages. We have clarified the 'contamination' outcomes so they closely reflect the wording in the questionnaire:

- reading someone else's messages in the trial (to be reported separately for intervention and control groups).
- someone else in the trial reading participants' messages (to be reported separately for intervention and control groups).

The reviewer asked if the outcomes could be placed in categories.

Thank you for your suggestion we have added the following categories of outcomes to improve clarity: STIs, behaviours, process outcomes, reading and sharing of the intervention, potential harms.

The reviewer asked about why the number of messages varied for people of different gender and sexual orientation.

Please see our response to the same query raised by reviewer 2.

The reviewer asked for more information about the control group messages, their frequency, topic and whether they are tailored.

We can confirm that the control group messages are not sent at the same frequency as the intervention messages. They are sent monthly asking for information about changes in postal or email addresses. For clarity, we have now added that they are not tailored and include an example control group message: 'Thank you for taking part in the texting study. Remember to let us know if your contact details have changed by replying to this text or emailing safetxt@lshtm.ac.uk'

Data analysis: Reviewer 3 asked for more information about the predictors of loss to follow up, predictors of outcome and whether they are the same co-variables.

Based on our experience in previous trials the predictors of outcomes can be, but are not always the same as, the predictors of loss to follow up. Not all the variables collected at baseline will predict both outcome and loss to follow up. Please note that (i) variables which are predictors of what the actual missing outcome values were, will always useful to include in an imputation model, as they recover information; (ii) where these variables also predict the chance of the outcome data being missing, they also reduce any bias that may be caused by the missing values. However, variables that only predict the chance of outcome data being missing do not recover any information and should not be used.(4).

Also, by principle, the number of imputations (in MICE) should be based on the percentage of missingness, and it gains no more precision to impute 100 datasets when only 15% of data are missing (15 is probably enough).

While for most analyses 15 imputations is fine, if we want to be confident our results are replicable to the number of decimal places given (both for standard errors and p-values a larger number of imputations is preferable (5)). As imputations are 'cheap' it seems better to be on the safe side, particularly as we do not currently know the proportion of missing outcome data.

References

1. Free C, McCarthy O, French RS, Wellings K, Michie S, Roberts I, Devries K, Rathod S, Bailey J, Syred J, Edwards P, Hart G, Palmer M, Baraitser P. Can text messages increase safer sex behaviours

in young people? Intervention development and pilot randomised controlled trial. Health Technol Assess 2016;20(57)

2. McCarthy O, French R, Baraitser R, Roberts I, Rathod S, Devries K, Bailey J, Edwards P, Wellings K, Michie S, Free C. safetxt: a pilot randomised controlled trial of a mobile phone-based intervention to increase safer sex behaviours in young people. BMJ Open 2016;6: e013045. doi:10.1136/bmjopen-2016-013045

3. Torgerson D J. Contamination in trials: is cluster randomisation the answer? BMJ 2001;322:355

4. Spratt M, Carpenter J, Sterne J, Carlin J, Heron J, Henderson J, Tilling K, Strategies for Multiple Imputation in Longitudinal Studies, American Journal of Epidemiology, Volume 172, Issue 4, 15 August 2010, Pages 478–487, <https://doi.org/10.1093/aje/kwq137>

5. Carpenter J, Kenward M. Multiple Imputation and its application. Statistics in Practice. Wiley 2013. p54.

VERSION 2 – REVIEW

REVIEWER	Guangyu Tong Duke University School of Medicine Department of Psychiatry and Behavioral Sciences
REVIEW RETURNED	24-Nov-2019

GENERAL COMMENTS	The authors clarified some of my previous concerns in the revision while left some others unexplained. It would be great to have a response letter to see the authors' thoughts more clearly, especially for the reviewers' question of whether this protocol is adding anything new; the justification for the effectiveness of the design, etc. I do see the new version is improved, and, in addition to the unaddressed comments, below are some other comments. On p.15, the authors clarified the harmful results of text messaging without giving any citations. "Car accidents are the only demonstrated harm resulting from text messaging, hence the intention to collect data regarding involvement in road traffic accidents. The safetxt intervention aims to increase partner notification of STI status. Fear of partner violence has been reported to be a barrier to partner notification and partner notification has been identified as a factor which may trigger partner violence." I am still not convinced by the intervention that is tailored for people to have different sexual orientations and gender. How could the recruiters be confident what the participants reported were accurate? And why are such tailored messages meaningful? (The STDs under study have different ways of transmission/protection methods for different groups?) Also, on p.9, it is better to explain what WSM, MSW, MSMW, etc, mean before using the acronyms.
--

VERSION 2 – AUTHOR RESPONSE

We would like to thank you and reviewers for their comments and time taken to review the protocol. Thank you for the opportunity to respond to reviewers comments.

The editor asked that we ensure that we have discuss the fact that the control group are not attention matched.

Under the control group description in the methods section we have added 'The control group is not attention matched as during the pilot work young people reported that it was irritating to receive the same number of messages as the intervention group about another health topic, when they had just been diagnosed with an STI. If the trial shows a benefit it is possible that this could be due to simply receiving the number of messages involved in the intervention, rather than the message content. However, the lack of effect of some mobile phone message based interventions in sexual health makes this less likely (14).'

The reviewer asked to see our response to the reviewers comments from Nov 2019
Thank you we have uploaded our response to the reviewers comments for the November submission.

The reviewer asked for us to give citations for the potential harms of text messaging.

We have included citations for these:

For road traffic accidents: Rothman KJ. Epidemiological evidence on health risks of cellular telephones. *Lancet* 2000; 356: 1837-40

for fear of partner violence for partner notification: Decker MR, Miller E, McCauley HL, Tancredi DJ, Levenson RR, Waldman J, Schoenwald P, Silverman JG. Intimate partner violence and partner notification of sexually transmitted infections among adolescent and young adult family planning clinic patients. *Int J STD AIDS*. 2011 Jun;22(6):345-7. doi: 10.1258/ijsa.2011.010425. PMID: 21680673; PMCID: PMC3603561.

"I am still not convinced by the intervention that is tailored for people to have different sexual orientations and gender."

The intervention is tailored in the sense that the precise set of messages users were served depended on their self-reported sex, self-report regarding whether or not they had a penis and the gender of their sexual partners (we made assumptions about anatomy based on sex and gender). We recognised that not all messages were relevant to all sex/ sexual orientation combinations. People with and without a penis require different messages, for example, about condoms. People with only same-sex sexual partners do not require messages about contraception.

"How could the recruiters be confident what the participants reported were accurate?"

We cannot be 100% confident that self-reports are accurate, a problem facing public health research in all areas where direct observation or measurement is not feasible. However, we can be moderately confident that participants' reports were on the whole accurate as we: conducted the discussion in a confidential community space not aligned to the research; used experienced, non-judgmental interviewers skilled in evoking sensitive topics; ensured respondents knew that the interviewer had not been involved in the intervention development (and so was not invested in it); employed techniques to facilitate disclosure of both negative and positive findings (Lee R. *Doing research on sensitive topics*. Sage London 1999); specifically asked for negative perceptions during the discussion if they did not arise.

Ultimately, the rationale for the trial is to establish the intervention effect on an object health outcome (Chlamydia or gonorrhoea infection within 12 months) so we can determine the effectiveness of the intervention without relying on self-report.

“And why are such tailored messages meaningful? (The STDs under study have different ways of transmission/protection methods for different groups?)”

We provided people with information specific to the STD they were diagnosed with. People face different sexual health risks and different behavioural challenges to reduce harm dependent on their anatomy and the anatomy of their sexual partners. It is essential to the users' perception of the relevance of the intervention that messages are appropriate to the users anatomy and the anatomy of their sexual partners.

Also, on p.9, it is better to explain what WSM, MSW, MSMW, etc, mean before using the acronyms. Thank you we have explained the acronyms before we use them.